

# TET1 may contribute to hypoxia-induced epithelial to mesenchymal transition of endometrial epithelial cells in endometriosis

Jingni Wu, Xidie Li, Hongyan Huang, Xiaomeng Xia, Mengmeng Zhang and Xiaoling Fang

Department of Obstetrics and Gynecology, The Second Xiangya Hospital, Central South University, Changsha, Hunan, China

## ABSTRACT

**Background**. Endometriosis (EMs) is a non-malignant gynecological disease, whose pathogenesis remains to be clarified. Recent studies have found that hypoxia induces epithelial-mesenchymal transition (EMT) as well as epigenetic modification in EMs. However, the relationship between EMT and demethylation modification under hypoxia status in EMs remains unknown.

**Methods**. The expression of N-cadherin, E-cadherin and TET1 in normal endometria, eutopic endometria and ovarian endometriomas was assessed by immunohistochemistry and immunofluorescence double staining. 5-hmC was detected by fluorescence-based ELISA kit using a specific 5-hmC antibody. Overexpression and inhibition of TET1 or hypoxia-inducible factor $2\alpha$ (HIF-$2\alpha$) were performed by plasmid and siRNA transfection. The expression of HIF-$2\alpha$, TET1 and EMT markers in Ishikawa (ISK) cells (widely used as endometrial epithelial cells) was evaluated by western blotting. The interaction of HIF-$2\alpha$ and TET1 was analyzed by chromatin immunoprecipitation.

**Results**. Demethylation enzyme TET1 (ten-eleven translocation1) was elevated in glandular epithelium of ovarian endometrioma, along with the activation of EMT (increased expression of N-cadherin, and decreased expression of E-cadherin) and global increase of epigenetic modification marker 5-hmC(5-hydroxymethylcytosine). Besides, endometriosis lesions had more TET1 and N-cadherin co-localized cells. Further study showed that ISK cells exhibited enhanced EMT, and increased expression of TET1 and HIF-$2\alpha$ under hypoxic condition. Hypoxia-induced EMT was partly regulated by TET1 and HIF-$2\alpha$. HIF-$2\alpha$ inhibition mitigated TET1 expression changes provoked by hypoxia.

**Conclusions**. Hypoxia induces the expression of TET1 regulated by HIF-$2\alpha$, thus may promote EMT in endometriosis.

## INTRODUCTION

Endometriosis is a chronic and non-malignant gynecological disease characterized by the growth of endometrial glands and stroma outside the uterus (*Giudice, 2010*). It exhibits

Corresponding author
Xiaoling Fang, fxlfxl0510@csu.edu.cn

cancer-like features, such as cell proliferation and metastatic invasion. Endometriosis is a major contributor to pelvic pain and infertility (*Mahmood & Templeton, 1991*). Although the etiology of endometriosis is still unclear, retrograde menstrual reflux is the widely accepted hypothesis for the mechanism of endometriosis. This theory states that retrograded endometrial tissues must migrate, invade and survive outside the cavity of uterus, then establish new endometriosis lesions. However, little is known about the molecular events that lead to the development of endometriosis.

Due to lack of hormone and blood supply, the retrograded endometrial debris during menstruation is in hypoxic status (*Maybin & Critchley, 2015*). Hypoxia plays a role in the migration and invasive of endometrial epithelial cells in endometriosis (*Xiong et al., 2016*). It probably induces the survival of retrograded endometrial debris and angiogenesis in implanted ectopic endometrial lesions (*Wu et al., 2007*). That is, the hypoxia microenvironment may contribute to the migration, invasion, and ectopic implant formation of the eutopic endometrial epithelial cells. Therefore, hypoxia has been regarded as an important stimulus of the pathological process of endometriosis. Hypoxia can stabilize hypoxia-inducible factors (HIFs, including HIF-1α and HIF-2α), which are the most significant and sensitive mediators of hypoxia-induced cellular responses. Stabilized HIFs dimerize with their constitutively stabilized partner, HIF-1β, and regulate the expression of the target gene, thereby leading to hypoxia-induced phenotypes (*Hsiao, 2015*; *Jain et al., 2018*). However, few studies have investigated the role of the transcription factor HIF-2α in endometriosis.

Studies have shown that hypoxia promotes the epithelial to mesenchymal transition (EMT) and enhances endometrial cell migration and invasion in endometriosis (*Liu et al., 2017*; *Liu et al., 2018*). EMT is a process by which epithelial cells lose polarity and cell-to-cell contacts and transform into mesenchymal cells with high motility. This process is characterized by the downregulation of the epithelial marker, E-cadherin, and the upregulation of mesenchymal markers, N-cadherin and vimentin (*Lamouille, Xu & Derynck, 2014*), which may occur in EMs (*Matsuzaki & Darcha, 2012*). EMT endows cells with migratory and invasive properties, a prerequisite for the establishment of endometriotic lesions (*Matsuzaki & Darcha, 2012*; *Xiong et al., 2016*).

Epigenetic alterations of chromatin (including DNA methylation, histone modifications, and non-coding RNAs regulation) are proposed to facilitate EMT in many diseases. Aberrations in DNA methylation are associated with EMT and tumorigenesis under hypoxia conditions (*Camuzi et al., 2019*). However, in endometriosis, the relationship between EMT and DNA methylation under hypoxia status remains largely unknown. The dynamic balance between methylation and demethylation is crucial for various biological processes (*Jones, 2012*). The altered expression of the demethylation enzyme ten-eleven translocation (TET1) disrupts this balance (*Lorsbach et al., 2003*), leading to aberrant DNA methylation patterns, which is seen in many human diseases, such as cancer (*Baylin & Jones, 2011*). TET1 enzymes iteratively oxidize 5-methylcytosine (5-mC) to 5-hydroxymethylcytosine (5-hmC), thereby contributing to CpG island demethylation in specific gene promoters (*Wu & Zhang, 2017*). TET1 is involved in cell migration, differentiation, and oncogenesis (*Yang et al., 2015*). A recent study has shown that TET

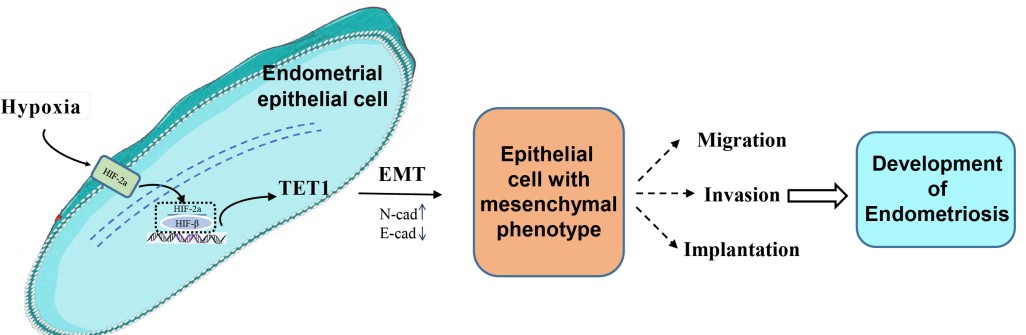

**Figure 1** **An infographic of this work.** Hypoxia may increase TET1 expression, mediated by HIF-2 $\alpha$, thus potentially inducing the EMT of endometrial epithelial cells and contributing to the development of endometriosis.

genes are dysregulated in endometriosis (*Roca et al., 2016*). However, the function of TET1 in endometriosis is not yet fully understood.

Our study investigated for the first time the role and mechanism of TET1 in regulating the EMT process of endometrial epithelial cells under hypoxic conditions. We hypothesized that hypoxia may increase TET1 expression, mediated by HIF-2α, thus potentially inducing the EMT of endometrial epithelial cells and contributing to the development of endometriosis. An infographic of this work is shown in Fig. 1.

# MATERIAL AND METHODS

## Patients and sample collection

The patients recruited in the study were women of childbearing age from the Second Xiangya Hospital. Patients with any hormonal-dependent disease, irregular menstrual cycles and those who took steroids and GnRH agonists for the past six months were excluded. The endometrial samples were obtained from these patients and were all confirmed in the early proliferation stage by pathological examination. This study was approved by the Human Ethics Committee of Second Xiangya Hospital, Central South University. Written informed consent was obtained from all the patients (Ref. No. 2016-243).

Different types of endometriotic lesions may have different pathogeneses (*Nisolle & Donnez, 1997*). Ovarian endometriosis lesions are more correlated with hypoxia and angiogenic factors (*Filippi et al., 2016*). Hence, we focused on the ovarian endometrioma in this study. Fifteen ovarian endometriomas and fifteen eutopic endometria (from the same group of women with ovarian endometriosis), and fifteen samples of normal endometria were used for the immunohistochemistry. The stages of endometriosis were classified according to the rASRM classification criteria during the procedure of laparoscopy. Clinical characteristics of the patients are shown in Table 1. Besides, these samples were fixed in 10% buffered formalin and embedded in paraffin for immunohistochemistry analysis.

**Table 1** Clinical characteristics of patients.

| | Immunohistochemistry samples | | |
| --- | --- | --- | --- |
| | **Normal endometria** | **Eutopic endometria** | **Ectopic endometria** |
| case number | 20 | 15 | 15 |
| *age | 35(27–41) | 29(27-38) | 29(27-38) |
| menstrual cycle phase | Early proliferation | Early proliferation | Early proliferation |
| #rASRM stage | | | |
| III | – | 9 | 9 |
| IV | – | 6 | 6 |

**Notes.**
Eutopic endometria were paired with ectopic endometria.
*Median (interquartile range).
# rASRM (Revised American Society for Reproductive Medicine classification, 1997).

## Immunohistochemistry (IHC)

Immunohistochemistry staining was performed on paraffin-embedded blocks of eutopic endometria, ovarian endometriomas and normal endometria. 5-$\mu$m-thick tissue sections were deparaffinized in xylene and rehydrated in graded ethanol, and antigen retrieval was performed in 0.01mol/l sodium citrate buffer (pH 6.0) using a pressure cooker. Sections were then treated with 3% hydrogen peroxide for 10 min to block endogenous peroxidase. After rinsing in PBST, sections were blocked in 5% bovine serum albumin for 30 min, then incubated overnight at 4 °C with primary antibodies. The second day, sections were incubated with horseradish peroxidase (HRP)-conjugated secondary anti-rabbit IgG for 30 min after rinsing for three times. Antigen-antibody reactions were detected by the HRP-catalyzed reaction with 3,3′-diaminobenzidine (DAB) (Beyotime Biotechnology, Shanghai, China). Finally, the sections were counterstained with hematoxylin and analyzed by optical microscopy. Tyramide signal amplification (TSA) technique was used to enhance the sensitivity of IHC. The following primary antibodies were applied: N-cadherin (# 66219-1-AP, proteintech, Wuhan, China, 1:1000), E-cadherin (#20874-1-AP, proteintech, Wuhan, China, 1:2000) and TET1 (ab191698, Abcam, Cambridge, UK, 1:1000).

## DNA isolation and determination of global 5-hmC levels

Genomic DNA from human endometrium and endometriosis lesions were isolated and purified using the CWBIO Genomic DNA kit according to the manufacturer's instructions. The concentration and quality of the DNA were estimated by agarose electrophoresis and UV absorption analysis. Global 5-hmC levels of this isolated genomic DNA were determined with a fluorescence-based Enzyme Linked Immunosorbent Assay (ELISA) kit using a specific 5-hmC antibody (Epigentek, Farmingdale, NY, USA), according to the manufacturer's recommendations. 200 ng of total genomic DNA was applied for this assay. The 5-hmC quantities in these DNA samples were analyzed based on the OD value generated by a microplate reader compared to control DNA in the kit. All samples were run in duplicate. These data were obtained from six normal endometria and six eutopic endometria and six ovarian endometriomas.

## Immunofluorescence double staining

Paraffin-embedded sections were deparaffinised, rehydrated, heated in 0.01 mol/l sodium citrate buffer (pH 6.0) for antigen retrieval, and treated with 3% hydrogen peroxide for 10 min to block endogenous peroxidase activity. After washing with PBS three times and blocking in 5% bovine serum albumin for 30 min, the sections were incubated in the primary antibody solution containing the rabbit anti-TET1 antibody (1:1000), rabbit anti-N-Cadherin antibody (1:200), and rabbit anti-E-cadherin antibody (1:50). Because we used TSA technique to enhance the fluorescent signal, antibodies of the same species were acceptable in our experiment. The biotinylated secondary anti-rabbit antibody was used to detect primary antibodies. The detection was performed with Streptavidin-HRP D (DAB Map kit, Ventana Medical Systems), followed by incubation with 488 (cat# T20922) or 594 (cat# T20935) Tyramide Alexa Fluors (Invitrogen) prepared in the dark according to the manufacturer's instructions. Sections were counterstained with DAPI.

## Image analysis

Immunohistochemistry and immunofluorescence double staining experiments were analyzed with a PerkinElmer Quantitative Pathology Imaging System in the Hunan Epigenetics Laboratory. This imaging system includes the Mantra Quantitative Pathology Workstation (PerkinElmer, CLS140089) for collecting staining data and PerkinElmer inForm software (PerkinElmer, Hopkinton, MA, USA) for analysis. The scanned images were visually examined. The samples with staining artifacts or with low quality were excluded. Image processing comprised the training session and analysis session. InForm software was trained to identify regions of interest and used to analyze the whole image. Three fields of view were randomly selected per slide to calculate the expression levels and distribution of TET1, E-cadherin, and N-cadherin in the glandular epithelium. The proportion of positive cells is calculated as the number of positive-staining cells divided by the number of total glandular epithelial cells. The correlation between TET1 and EMT markers expression in the endometrial epithelium (positive cell proportion) was analyzed by Pearson correlation test.

## Cell culture and hypoxia treatment

Ishikawa (ISK) cells (ZQ0472, Zhong Qiao Xin Zhou Biotechnology Co., Ltd, Shanghai, China) come from well-differentiated human endometrial adenocarcinoma. The human endometrial epithelial cell is hard to passage and transfect; endometrial stromal cells are the only cells to survive after 3–4 generations *in vitro*. Therefore, ISK cells are widely used instead of human endometrial glandular epithelial cells in studies of endometriosis (*Guay & Akoum, 2007*; *Cho et al., 2016*). ISK cells were cultured in Dulbecco's Modified Eagle's Medium/Nutrient Mixture F-12 (DMEM/F12) supplemented with 10% fetal bovine serum (FBS) and 1% penicillin-streptomycin in 5% $CO_2$ incubators at 37 °C until 70% confluence was reached. To ensure adequate nutrients and growth factors, the medium was renewed 1 h prior to hypoxia treatment. Thereafter, ISK cells were cultured under hypoxia (5% $O_2$) or normoxia (20% $O_2$) conditions for 4, 8, and 24 h.

## Gene transfection

ISK cells were seeded in six-well plates for 24 h to reach approximately 70% confluence and transfected with TET1 or the negative control plasmid (GenScript Biotech Corporation; Piscataway, NJ, USA) using Lipofectamine (Invitrogen, Carlsbad, CA, USA), according to the manufacturer's protocol. In Brief, we used the QIAGEN$^R$ kit to purify and extract the plasmid. Then, 1–4 μg of plasmid DNA and Lipofectamine solution were mixed in serum-reduced medium and added to the ISK cells. After 6 h of incubation, fresh medium was added. For TET1 downregulation, TET1-targeting siRNAs and the negative control (NC) were obtained from Ribobio (Guangzhou, China). The RNA interference (RNAi) experiments were performed following the manufacturer's protocol.First, the TET1 siRNA and NC siRNAs were each mixed with RNA transfection buffer (Ribo FECT CP) and then added to ISK cell medium. After 48 h, western blotting was performed to determine the transfection efficiency.

## Protein extraction and western blotting analysis

Total proteins were extracted from cells using RIPA lysis buffer and phosphatase inhibitors. All lysates were centrifuged at $12,000\times$ g for 20 min. The supernatants were collected, and the protein concentrations were determined by the Bicinchonic Acid (BCA) protein assay kit (Beyotime Biotechnology, Shanghai, China). Proteins were mixed with the same amount of loading buffer and boiled for 5 min. The samples were then resolved by Sodium dodecyl-sulfate polyacrylamide gel electrophoresis (SDS-PAGE), transferred to PVDF membranes, incubated with 5% fat-free milk, and hatched with the following primary antibodies at 4 °C overnight: N-cadherin, E-cadherin, TET1, HIF-2α, and HIF-1α (Abcam, Cambridge, UK). The membranes were washed with TBST three times for 15 min, and then incubated in a secondary antibody at 37 °C for 1.5 h. The blots were analyzed by a chemiluminescence system (Millipore. USA). These experiments were repeated three times, and the average values of the blot bands were calculated.

## Co-immunoprecipitation

Co-immunoprecipitation was carried out to identify protein–protein interactions. Cells were collected after exposure to hypoxia. Cell pellets were lysed in RIPA lysis buffer containing phosphatase inhibitors. 750 μl of the cell lysates were incubated with 2 μg of antibodies (rabbit HIF-2α, mouse HIF-1α, and rabbit TET1) overnight at 4 °C. Meanwhile, β-actin, rabbit IgG, and mouse lgG (Proteintech, Wuhan, China) were used as the input and negative controls, respectively, for the experiments. On the second day, each group was mixed with 20 μl of Protein G Plus/Protein A Agarose Suspension (Santa Cruz Biotechnology, Santa Cruz, CA, USA) and and the solution was softly shook at 4 °C for 2 h. To separate agarose beads, these mixtures were centrifuged at 3,000 rpm for 3 min at 4 °C. The supernatant was transferred to a new tube on ice, and the beads were eluted again with lysis buffer. The total eluted supernatants were washed 5 times, resuspended in $1 \times$ loading buffer, and boiled for 5 min at 100 °C before western blotting.

## Statistical analysis

Statistical analysis of data was performed by one-way ANOVA using SPSS 22.0 software (SPSS Inc., Chicago, IL, USA). All experiments were repeated three times. The results are shown as the mean ± standard deviation (SD). A $P$ value of <0.05 was considered statistically significant.

# RESULTS

## Accompanied with TET 1 upregulation, EMT might occur in endometrial epithelial cells of ovarian endometriosis

To evaluate the expression and localization of TET1 and EMT markers, E-cadherin and N-cadherin, in endometriosis, we performed IHC and immunofluorescence double staining in human endometrial tissues. Since the morphology of stromal cells in the ovarian endometriomas is much different from that in the normal endometria and eutopic endometria, we only analyzed the glandular epithelial cells in the three groups.

The expression of TET1 and EMT markers in the glandular epithelium of normal endometria, eutopic endometria, and ovarian endometriomas was analyzed by IHC. Figs. 2A–2R show the representative IHC results. Table 2 illustrates the expression of TET1 and EMT markers. The proportions of cells positively stained for TET1 or N-cadherin in the glandular epithelium of the eutopic endometria and ovarian endometriomas were significantly higher than those in the normal endometria, and the number of E-cadherin–positive cells was significantly lower in the glandular epithelium from eutopic endometria and ovarian endometriomas than those in the normal endometria (Figs. 2S–2U). The normal ovary tissue has a low expression of TET1 (Fig. S1). Moreover, to demonstrate whether the demethylation enzyme TET1 was activated in endometriosis, we explored the expression of 5-hmC, which is an important marker for the activated demethylation process. Figure 2V shows that the ovarian endometriomas have higher 5-hmC levels (5.15 ± 1.488% of total DNA) than eutopic endometria (3.33 ± 0.876% of total DNA, $p < 0.05$) and normal endometria (3.5 ± 0.881% of total DNA, $p < 0.05$). All in all, these data suggest that TET1 is upregulated and EMT may occur in the glandular epithelium of endometriosis.

Immunofluorescence double staining was performed to compare the localization of TET1 and EMT markers. Consistent with the IHC results, in eutopic endometria and normal endometria, TET1 and N-cadherin were mainly expressed in the stromal cells and barely expressed in the glandular epithelial cells. However, in the ovarian endometriomas, TET1 and N-cadherin were both expressed in the glandular epithelial cells. Furthermore, the distribution of TET1 was more obvious in the cytoplasm or nuclear of N-cadherin–positive cells (Figs. 2FF–2NN), not E-cadherin–positive cells (Figs. 2W–2EE). Ovarian endometriotic lesions (Figs. 2LL–2NN) had more TET1 and N-cadherin co-localized cells compared to eutopic endometria (Figs. 2II–2KK) and normal endometria (Figs. 2FF–2HH). The large amount of co-localized cells suggests a close relationship between TET1 and N-cadherin. In addition, a correlation analysis was applied to the proportions of cells that were positively stained for TET1 and N-cadherin/E-cadherin in the epithelium (Figs. 2OO–2PP), which showed that TET1 was positively correlated with N-cadherin and

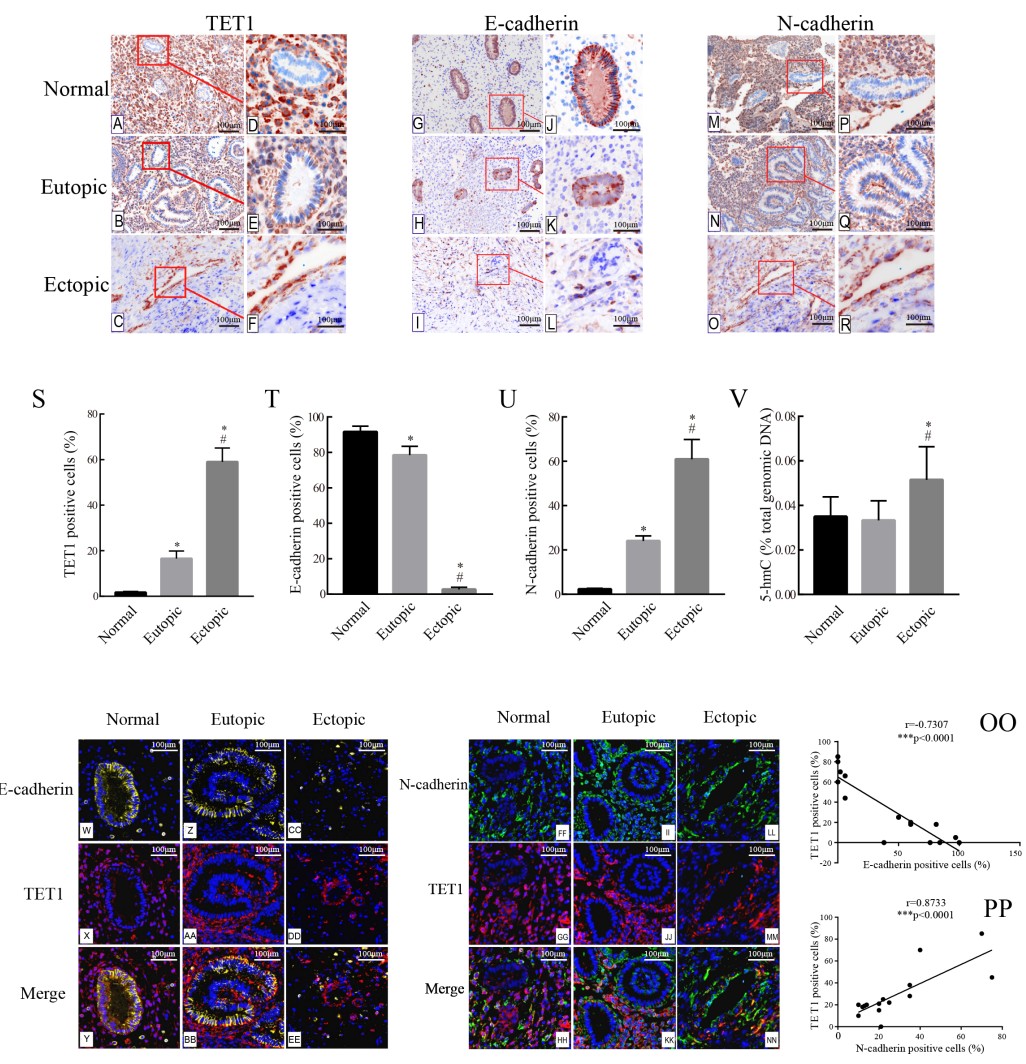

**Figure 2   Accompanied with the TET1 upregulation, EMT might occur in endometrial epithelial cells of ovarian endometriosis.** (A–R) Immunohistochemistry for TET1 and EMT markers (E-cadherin and N-cadherin) in epithelial gland cells of the normal endometria (Normal), eutopic endometria (Eutopic) and ovarian endometrioma (Ectopic). A, B, C, G, H I, M, N, O : Magnification ×100, D E, F, J, K, L, P, Q, R: Magnification ×200. (S–U) The proportions of cells positively stained for TET1, E-cadherin or N-cadherin in Normal, Eutopic and Ectopic groups (*$p < 0.05$ versus normal endometria; #$p < 0.05$ versus eutopic endometria). (V) Expression of global 5-hmC in the total genomic DNA (*$p < 0.05$ versus normal endometria; #$p < 0.05$ versus eutopic endometria). (W–EE) Representative double immunofluorescence images for TET1 and E-cadherin in the Normal (W–CC), Eutopic (Z–BB) and Ectopic (CC–EE) groups. (FF–NN) Representative double immunofluorescence images for TET1 and N-cadherin in the Normal (FF–HH), Eutopic (II–KK) and Ectopic (LL–NN) groups. (OO–PP) The correlation between TET1 and EMT markers expression in the endometrial epithelium is confirmed by quantitative analysis of the double staining immunofluorescence. Magnification ×200 (Left, red luminescence represents TET1. Yellow luminescence represents E-cadherin. Right, red luminescence represents TET1. Green luminescence represents N-cadherin). Normal, normal endometria; Eutopic, eutopic endometria; Ectopic, ovarian endometriomas.

**Table 2** Expression profiles of TET1 and EMT markers were detected by immunohistochemistry in the epithelium of normal endometria, eutopic endometria, and ectopic endometria.

| Marker | Positive epithelial cells for marker, % | | | | | |
|---|---|---|---|---|---|---|
| | Normal endometria, % (group1, $n = 15$) | Eutopic endometria, % (group2, $n = 15$) | Ectopic endometria, % (group3, $n = 15$) | *p*-value (1 versus 2) | *p*-value (2 versus 3) | *p*-value (1 versus 3) |
| E-cadherin | $91.66 \pm 3.196$ | $77.75 \pm 4.024$ | $2.668 \pm 1.174$ | <0.05[*] | <0.01[**] | <0.01[**] |
| N-cadherin | $2.235 \pm 0.339$ | $24.11 \pm 2.237$ | $60.98 \pm 8.858$ | <0.01[**] | <0.01[**] | <0.01[**] |
| TET1 | $1.668 \pm 0.438$ | $16.57 \pm 3.308$ | $59.03 \pm 6.042$ | <0.01[**] | <0.01[**] | <0.01[**] |

Notes.
[*] $P < 0.05$
[**] $P < 0.01$
All data are expressed as mean + SD.

negatively correlated with E-cadherin. Our findings suggest that the demethylation enzyme TET1 may play a role in the EMT pathological process of endometriosis.

## Hypoxia alters the expression of HIFs, TET1 and EMT markers

Although various epigenetic mechanisms have been demonstrated to regulate EMT under hypoxia status (*Wu et al., 2012*; *Tsai & Wu, 2014*), the role of the DNA demethylation enzyme TET1 in regulating hypoxia-induced EMT in endometriosis remains largely unknown. Therefore, we investigated the expression of TET1, HIFs (HIF-1 α and HIF-2α), and EMT markers (E-cadherin, N-cadherin, and vimentin) in ISK cells exposed to 5% $O_2$ for 0 h, 4 h, 8 h and 24 h. As shown in Fig. 3, the expression of TET1, HIF-1α, HIF-2α, and vimentin increased significantly and the expression of E-cadherin decreased significantly after hypoxia treatment for 8 h and 24 h, especially for 24 h. N-cadherin expression peaked at 8 h after hypoxic treatment and went down at 24 h, we presumed N-cadherin might be transiently triggered by hypoxia. Homeostasis should work to reverse the hypoxia condition, and genes may be activated in different time period of hypoxia during this process (*Deret et al., 2004*; *Javaid et al., 2013*; *Song et al., 2019*). Hence, the time point with the maximal response was selected, and vimentin was chosen as a mesenchymal marker in the following experiences. Our findings indicate that hypoxia activates TET1 expression and the EMT.

## TET1 overexpression may promote EMT, and knockdown of TET1 mitigates hypoxia-induced EMT in ISK cells

To evaluate the role of TET1 in hypoxia-induced endometrial EMT, we successfully established TET1-overexpressing ISK cells under normal conditions and TET1 down-regulation ISK cells under hypoxia conditions. As seen in Fig. 4, western blotting indicated that the expression of vimentin was increased and the expression of E-cadherin was decreased by upregulation of TET1 under normoxia. In addition, hypoxia-induced EMT, characterized by the repression of E-cadherin and the upregulation of vimentin, was abolished by TET1 knockdown. These results reveal that TET1 plays a crucial role in hypoxia-induced EMT.

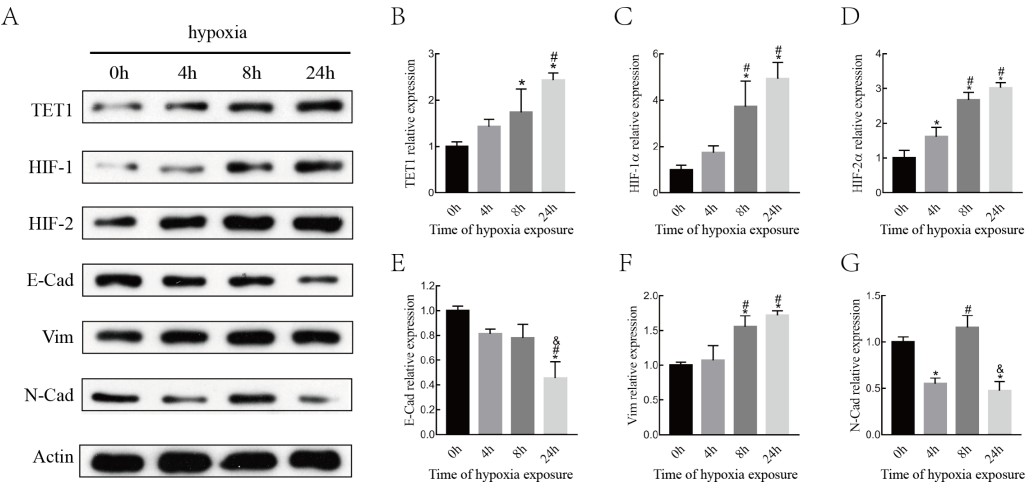

**Figure 3** **Hypoxia alters the expression of HIFs, TET1 and EMT markers.** (A–B) Western blotting analysis showed the expression of HIF-1$\alpha$, HIF-2$\alpha$, TET1 and EMT markers (E-Cad, E-cadherin; N-Cad, N-cadherin and Vim, Vimentin) in Ishikawa(ISK) cell cultured under hypoxia at different time point (0 h, 4 h, 8 h and 24 h) (*$p < 0.05$ versus normoxia; #$p < 0.05$ versus hypoxia for 4 h; & $p < 0.05$ versus hypoxia for 8 h).

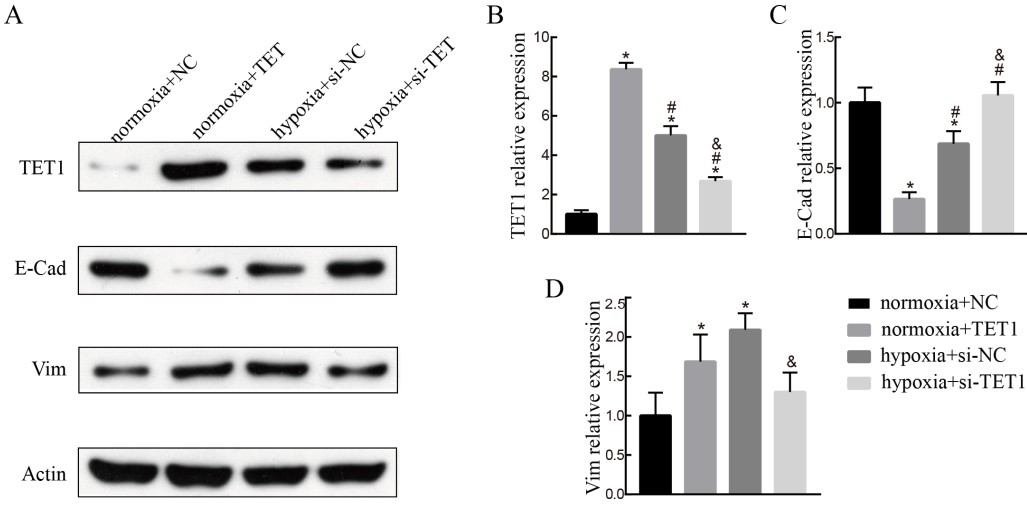

**Figure 4** **TET1 overexpression may promote EMT and knockdown of TET1 mitigates hypoxia-induced EMT.** (A, B) Western blotting analysis showed the expression of vimentin (Vim) and E-cadherin (E-Cad) in cells transfected with TET1 expression plasmid (TET1) or empty vectors (NC) under normoxia conditions, and cells transfected with TET small interfering RNA (si-TET1) or negative control (si-NC) under hypoxia conditions. (*$p < 0.05$ versus normoxia+NC; #$p < 0.05$ versus normoxia+TET1; & $p < 0.05$ versus hypoxia+si-NC).

## Knockdown of HIF-2$\alpha$ inhibits hypoxia-induced EMT and TET1 expression in ISK cells

To investigate the mechanism of TET1 in the regulation of hypoxia-induced EMT, hypoxia-induced HIF-2$\alpha$ expression was inhibited by a specific siRNA for 48 h. Western blotting

analysis confirmed the down-expression of HIF-2α. Knockdown of HIF-2α mitigated the activation of TET1 and EMT under hypoxia in ISK cells (Figs. 5A and 5C–5F), indicating that HIF-2α is a regulator of TET1 expression and the EMT under hypoxia. *Xiong et al. (2016)* reported that HIF-1α also induced EMT in endometrial epithelial cells under hypoxia. Therefore, we conducted chromatin immunoprecipitation experiments, which showed that the anti-TET1 antibody pulled down HIF-2α and TET1, not HIF-1α, and the anti-HIF-2α antibody pulled down both TET1 and HIF-2α. This implied that HIF-2α, not HIF-1α, is directly or indirectly bound to the TET1 protein (Fig. 5B). Reporter gene assay demonstrated that the promoter region of the TET1 gene was activated by hypoxia/HIF-2α (*Tsai et al., 2014*). All these results indicate that HIF-2α may serve as a transcription activator in ISK cells to regulate TET1 involving hypoxia-induced EMT.

## DISCUSSION

Aberrant DNA methylation may be associated with the pathogenesis of endometriosis (*Arosh et al., 2015*; *Koukoura, Sifakis & Spandidos, 2016*; *Li et al., 2017*; *Juanqing et al., 2019*). TET1-mediated DNA hydroxymethylation is a crucial mechanism of DNA demethylation (*Jeschke, Collignon & Fuks, 2016*). The balance between methylation and demethylation is important for gene expression and cellular environmental homeostasis (*Song & He, 2012*; *Yang et al., 2015*). However, the demethylation modification in endometriosis has not yet been well studied. Our study shows for the first time that the elevated expression of the demethylation enzyme TET1 may be associated with the activated EMT phenotype in the epithelia of endometriotic lesions, which may provide a novel insight into the pathogenesis of EMs.

Genome-wide DNA hypomethylation occurs in many diseases (*Ehrlich, 2009*; *Song & He, 2012*; *Lim et al., 2015*). 5-hmC is a critical epigenetic modification marker that plays a noteworthy role in regulating gene expression (*Lorsbach et al., 2003*; *Mariani et al., 2014*; *Jeschke, Collignon & Fuks, 2016*). Our study had some consistency with the report by *Roca et al. (2016)*. Although Roca et al. showed that TET1 was down-regulated in endometriotic tissues, the contribution of TET1 expression from each component of the endometrium (epithelium or stroma) is unknown. Endometriosis is an estrogen-dependent disease. Roca et al. found that global 5-hmC was upregulated in endometriotic tissues and TET1 was upregulated in endometrial epithelial cell line treated with estradiol, which supports our study and suggests that TET1 in epithelial cells may affect the expression levels of 5-hmC globally or locally. It is worth noting that the stromal cells are more loosely arranged in the ovarian endometriomas compared to the normal endometria and eutopic endometria. In addition, the endometriotic stromal and epithelial cells have different molecular expression profiles (*Logan, Yango & Tran, 2018*; *Noë et al., 2018*), and the endometriotic epithelial cells might originate from endometrial epithelial cells, whereas the origin of endometriotic stromal cells remains to be investigated (*Matsuzaki & Darcha, 2012*). Hence, we focused on the endometrial epithelium, which might play a specific and dynamic role in the pathogenesis of endometriosis (*Logan, Yango & Tran, 2018*; *Noë et al., 2018*). A recent study showed that decreased TET1 expression led to 5-hmC loss in ISK cells

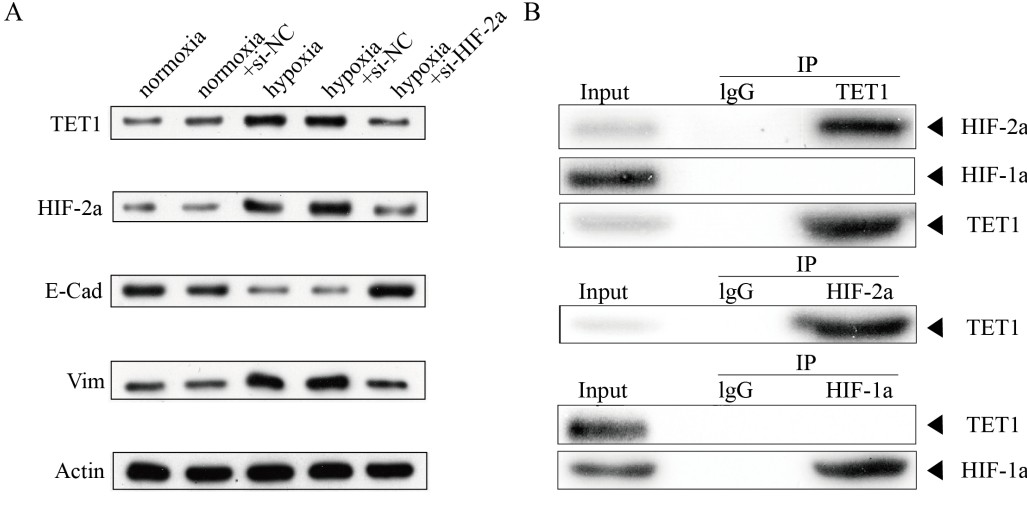

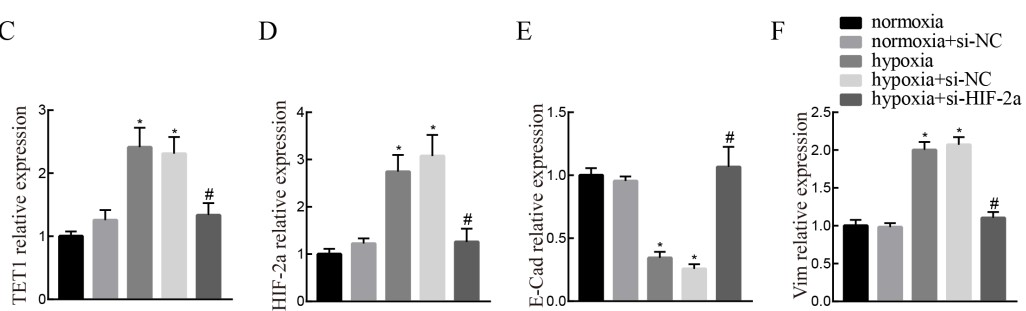

**Figure 5 Knockdown of HIF-2α inhibits hypoxia-induced EMT and TET1 expression.** (A, C) Western blotting showed the expression of TET1, HIF-2a, vimentin (Vim) and E-cadherin (E-Cad) in cells under normoxia conditions and cells transfected with HIF-2a small interfering RNA (si-HIF-2a) or negative control (si-NC) under hypoxia conditions ($*p < 0.05$ versus normoxia or normoxia+si-NC; $\#p < 0.05$ versus hypoxia or hypoxia+si-NC). (B) Co-immunoprecipitation of TET1, HIF-2α or HIF-1α. Hypoxia treated ISK cells were lysed and immunoprecied by TET1, HIF-2α, HIF-1α and normal rabbit/mouse lgG antibodies, respectively. Cell lysates (input) and immunoprecipitated proteins (IP) were analyzed by western blotting.

(*Lv et al., 2017*). We need to explore the distribution of 5hmC in endometriosis, especially in the epithelia, and the specific gene whose promoter is enriched with 5-hmC in future studies.

TET1 was mainly expressed in stromal cells but not epithelial cells in the normal and eutopic endometria, whereas the epithelial expression of TET1 expression was ubiquitous and stromal staining was scattered in ovarian endometriotic lesions. What's more, we showed a significant inverse expression between TET1 and the epithelial marker E-cadherin and a positive correlation between the expression of TET1 and the mesenchymal marker N-cadherin. Therefore, our study demonstrates for the first time that the enhanced epithelial expression of TET1 in ectopic lesions may play a crucial role in the EMT phenotype of endometriosis while the role of TET1 in stromal cells needs to be further investigated. Our

research may lay the foundation for this new demethylation mechanism of endometriosis. The localization of TET1 in both the nucleus and cytoplasm was surprising since TET1 is a nuclear protein. However, the similar subcellular localization of TET1 was also shown in prostate, gastric cancer, hippocampus neurons and so on (*Hsu et al., 2012*; *Kaas et al., 2013*; *Fu et al., 2014*; *Han et al., 2017*). TET genes localize in the cytoplasm of neurons to sustain cell survival (*Mi et al., 2015*) and localize in the cytoplasm of colorectal cancer cells to promote tumor metastasis (*Huang et al., 2016*). The variations of TET1 immunostaining results indicate a possible regulation of TET1 subcellular localization by still unclear signaling pathways.

Hypoxia is an important factor of the endometriosis microenvironment (*Becker et al., 2008*; *Wu, Hsiao & Tsai, 2019*). Hypoxia can induce epigenetic changes in tumor cells, whereas hypoxia-induced epigenetic changes in endometriosis haven't been reported yet (*Zhou et al., 2006*; *Shahrzad et al., 2007*). In our study, we showed for the first time that hypoxia/HIF-2α activated TET1 expression, and TET1 knockdown mitigated hypoxia-induced EMT in ISK cells. Co-immunoprecipitation revealed that HIF-2α, not HIF-1α, was bound to the TET1 protein in ISK cells. Therefore, we did not further explore the role of HIF-1α in our study. This co-immunoprecipitation result of TET1 and HIF-1a is different from the data reported by *Tsai et al. (2014)* and *Cheng et al. (2018)*. Tsai et al. showed the interaction of TET1 and HIF-1a based on the 293 T cell line while Cheng et al. showed the same interaction based on the cells in mouse prefrontal cortex. Whereas, our study was performed on the ISK cells which could have different protein-protein interactions. What's more, *Tsai et al. (2014)* concluded that HIF-2α was the major regulator of TET1 expression under hypoxia. In this study, the HIF-2a knockdown didn't eliminate the expression of TET1, which might indicate the existence of other regulators. *Hu et al. (2018)* reported microRNA-210 was involved in the regulation of TET1 under hypoxia status. *Lin et al. (2017)* showed HIF-1a could regulate the hypoxia-induced expression of TET in hepatoblastoma HepG2 cells, whereas *Tsai et al. (2014)* showed the HIF-1a knockdown had no effect on the TET1 expression in cancer cell lines. It will be explored in future studies. Combined with the reporter gene assay of HIF-2α and TET1 (*Tsai et al., 2014*), we speculate that the hypoxia microenvironment may induce the EMT of endometrial epithelial cells via the activation of TET1 partly regulated by HIF-2α. These epithelial cells may pass through oviducts and implant into the ovarian surfaces, thus contributing to the development of endometriosis.

Our study had several limitations. First, we used ISK cells for *in vitro* experiments instead of the primary endometrial epithelial cells. The primary endometrial epithelial cells could hardly survive in primary culture. Therefore, ISK cells were used in numerous studies of endometriosis (*Cho et al., 2016*; *Lee et al., 2018*; *Matson et al., 2018*; *Choi et al., 2018*). The ISK cell is a well-differentiated human endometrial adenocarcinoma cell line. It retains the phenotype of endometrial epithelial cells, bears estrogen and progesterone receptors, and displays a similar molecular expression profile as endometrium (*Du et al., 2018*). However, our study will be more convincing if we use more cell lines. With the advancement of technology and improvement of experimental approaches, a deep understanding of the roles of TET1 in endometriosis will be possible. Secondly, the precise role and mechanism

of TET1 in regulating the EMT remain to be investigated. The TET1 target gene and the correlation of TET1 with the EMT transcriptional factors slug, snails, and twist need to be elucidated in future studies. The interaction between TET1 and HIF-2α needs to be demonstrated by additional experiments, such as GST pull-down and yeast two-hybrid experiments. Thirdly, although 5-hmC detection is very expensive, analyzing more samples for overall 5-hmC expression in the epithelia and identifying the specific gene enriched with 5-hmC are still needed to study the pathogenesis of endometriosis.

## CONCLUSION

To our knowledge, we are the first to explore the relationship between EMT and the demethylation enzyme TET1 in endometriosis. Hypoxia induces the expression of TET1, mediated by transcription factor HIF-2α, which may promote the EMT of endometriosis. These data provide a new understanding of the pathological process of endometriosis, which may advance knowledge of the epigenetic mechanism as well as the therapeutic approach towards endometriosis.

### Funding

This work was supported by the National Natural Science Foundation of China (grant number 81671437, 81771558), the Natural Science Foundation of Hunan Province (grant number 2020JJ4814), and the Postgraduate Independent Exploration and Innovation Project of Central South University, China (grant number 2020zzts285). The funders had no role in study design, data collection and analysis, decision to publish, or preparation of the manuscript.

### Grant Disclosures

The following grant information was disclosed by the authors:
National Natural Science Foundation of China: 81671437, 81771558.
Natural Science Foundation of Hunan Province: 2020JJ4814.
Postgraduate Independent Exploration and Innovation Project of Central South University, China: 2020zzts285.

### Competing Interests

The authors declare there are no competing interests.

### Author Contributions

- Jingni Wu conceived and designed the experiments, performed the experiments, analyzed the data, prepared figures and/or tables, authored or reviewed drafts of the paper, and approved the final draft.
- Xidie Li performed the experiments, analyzed the data, prepared figures and/or tables, authored or reviewed drafts of the paper, and approved the final draft.
- Hongyan Huang performed the experiments, prepared figures and/or tables, and approved the final draft.

- Xiaomeng Xia analyzed the data, authored or reviewed drafts of the paper, and approved the final draft.
- Mengmeng Zhang analyzed the data, prepared figures and/or tables, and approved the final draft.
- Xiaoling Fang conceived and designed the experiments, prepared figures and/or tables, authored or reviewed drafts of the paper, and approved the final draft.

## Human Ethics

The following information was supplied relating to ethical approvals (i.e., approving body and any reference numbers):

This study was approved by the Human Ethics Committee of Second Xiangya Hospital, Central South University (Ref. No. 2016-243).

Written informed consent was obtained from all the patients. Our project about the epithelial-mesenchymal transition regulation in endometriosis conforms to the principles and moral requirements of medical ethics.This study belongs to experimental study, intervention study and theoretical research.The specimen types includes the pathological tissue, individual patient data and cell line.

## Data Availability

The raw measurements are available as Supplemental Files.

## Supplemental Information

Supplemental information for this article can be found online at http://dx.doi.org/10.7717/peerj.9950#supplemental-information.

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
