# Peer review of "TET1 may contribute to hypoxia-induced epithelial to mesenchymal transition of endometrial epithelial cells in endometriosis"

_PeerJ, doi:10.7717/peerj.9950_

## Round 0.1 · original submission · Major Revisions

I highlight a couple of things. It is important that your data and the interpretations you make of them be sound and convincing to your peers. Please keep in mind that you should include all relevant controls - of the siRNA and other reagents used (eg. more than one siRNA, a scrambled control not showing knockdown effect; stains without primary antibody), and of changing oxygen conditions (eg. return to normoxia for Figure 4) - to best make your points. I urge you to consider how you might "link the mechanism from endometrial epithelial cells back to the endometriosis tissue" as one reviewer suggested. All agree that the figures need significant revision, and that the language is not adequate.

Journal guidelines are that you should use clear and unambiguous phrases that are grammatically correct and conform to professional standards of courtesy and expression. I would urge you to have your revised manuscript reviewed by a qualified professional science translation service, of which there are now many. Even if the reviewers' written English itself is not perfect, they are not trying to publish their comments. If you resubmit without having made the language publication-ready, I will decide to reject the manuscript. PeerJ does not offer copyediting as standard and reviewers should not have to provide that service in your stead.

We do appreciate your contribution and hope that you will be able to make the major revisions requested and others, if you so choose.

Reviewer 1 ·

Basic reporting

1. Basic reporting. Authors explore the role of TET1 with in the frame of hypoxia and endometriosis pathogenesis. The body of this work aims to show hypoxia induces increased TET1 expression through HIF2a which increases 5-mhc driving increase EMT.

In general the language was satisfactory, a strong proofreading will be helpful for the final round of edits.
Examples include – consistent spacing between end of text and citations
- Line 45: debris – not debrises
- Line 244: N-cadherin – not N-cadherine
Introduction was written in a way that minimized previously published research by using language such as: could and may. These works have been published and are the ideas of those authors.
Transitions within the introduction between topics will significantly aid in the flow and readability.
TET1, TET2, and TET3 were previously characterized in human samples of endometriosis this work should be acknowledged within the introduction or the strong language of “the function of TET1 has not yet been well characterized” should be altered.
Figure 1 - ectopic images are very hard to see higher quality photomicrographs are necessary for readers to come to the same conclusions as authors. Within Figure 1D there is a floating scale bar in the middle of the image. And the labelling in 1D is not explained or referenced anywhere the 1Da,b,c,d,-i.
Figure 2 – symbols used within the figure need to be defined in the figure legend *, #, &.
N’s should be included in each figure legend
Table 1 – it needs to be indicated somewhere within the table that eutopic and ectopic endometria are matched samples
Table 3 – does not provide a significant amount of information and the data can be easily written into the text of the results.

Experimental design

2. Experimental design. Authors used protein analysis through IHC and Western blots to visualize TET1 and EMT in endometriosis. They then used a cell line to look at hypoxia and EMT markers in endometrial tissue. Finally, authors linked TET1 to Hif2a with a pull down assay.
Research question – Authors are attempting to explain the pathogenesis of endometriosis. EMT has long been recognized as a player in endometriosis and may publications have discussed the role of hypoxia in the initiation of EMT. There have additionally been a number of players that have been linked to the hypoxia induced EMT: Hif1a, TAF1, Perostin. Thus the knowledge gap is unclear.
Within the methods antibodies were cited but concentrations were not, please add.
It is unclear why only Ishikawa cells were used. Ishikawa cells are an endometrial cancer cell line which most likely carries a very different profile than the non-malignant endometriosis cells. There are a few endometriosis epithelial cell lines available that may have been better suited for these experiments. It is understandable that one is trying to show cells from the uterus become endometriotic cells but using only one cell line is not sufficient to represent this data.

Validity of the findings

3. Validity of the findings. Authors showed an increase in TET1 expression in ectopic lesions compared to control and eutopic endometrium. TET1 increase was correlated with changes in cadherins. Hypoxia was shown to be related to TET1 changes.

The use of human biopsies made the exploration of TET1 expression highly translatable.

It is unclear what the role of cell type localization of TET1 plays in this phenotype. Both the normal and the eutopic endometrium express TET1 within the stroma while the ectopic endometrium shows epithelial expression. This divergence is seemingly imperative to the story and needs significant expounding. Additionally, what is the expression of TET1 in normal ovaries specifically the epithelium. An additional panel of normal ovary would be beneficial.

The antibody used is suggested to be nuclear therefore an explanation as to why the staining appears cytoplasmic or images which clarify this is necessary.

Figure 1Ac you are unable to discern the endometriotic lesion being highlighted. In fact, all of these lesions are difficult to discern as endometriosis as no underlying stroma appears to be present.

Figure 1D. Again the lesion is very difficult to depict. The TET1 expression appears to be in the stroma and not epithelial which contradicts what was seen in 1A.

A clear discussion on why these results significantly differ from that found in Roca 2016 are necessary.

What controls were in place to assure the siRNA knockdown worked? The siTET does not appear to rescue hypoxia induced TET1 expression.

Figure 4. needs to have normaxia controls. Readers are unable to determine if the reduction is biologically relevant without seeing if the alterations cause a return to normal. Additionally, while there was a reduction in TET1 after siHIF-1a it was not eliminated, thus a discussion on other players is necessary.

A clear discussion on why these results differ from previously published with respect to TET1 and HIF1a are necessary. Both Tsai 2014 and Cheng 2018 show direct interact of TET1 and HIF1a. With Tsai showing it’s role in hypoxia induced EMT.

The in vitro experiments do not represent endometriosis and if they authors were able to link the mechanism from endometrial epithelial cells back to the endometriosis tissue it would strengthen the work. Do the ectopic lesions express HIF1a, HIF2a or Vim?

Additional comments

An infographic explaining the relationship between all presented parties and how they interact to promote disease would be highly beneficial.

Reviewer 2 ·

Basic reporting

The manuscript by Wu et al study the role of TET1, a DNA methylase in regulating hypoxia induced EMT in endometriosis. The current conclusions are of sufficient novelty and breadth of interest to merit publication in PeerJ. However, English should be checked and refined as needed throughout the manuscript. Here are few of the examples:

Line 57. Please correct transcript factor to transcription factor
Line 124. Please correct table number 2. Global 5-hmC levels are shown in table 3.
Line 244. Please correct N-cadherine to N-cadherin
Line 250. Please correct down-expressed to down-regulation
Line 282. Please correct transcript factor to transcription factor
Line 297, 288, 305, 306. Please correct the sentences.
Line 299. Please correct Hyper-methylation to Hypermethylation

Experimental design

No comment.

Validity of the findings

For Figure 2 A and B.

1) Authors reported N-cadherin expression peaks at 8 hour during hypoxia conditions. However, surprisingly the levels go down drastically at 24 hours (See Figure 2A and B). This is contradictory and authors need to clarify or comment on the expression pattern.

2) Please explain the meaning of symbols * and # in figure legends

Additional comments

Please be consistent with symbols throughout the manuscript. For example, hours in manuscript are sometime refered as h or sometime as hr.

Reviewer 3 ·

Basic reporting

1. The authors should check that all the labels of figures and tables in this article are correctly ordered. For example, in line 124, table 2 is not correct.
2. The description of Figure legends should be concise and pointed. For instance, The description of Figure 3 is too lengthy and to some extent similar to methods. What’s more, the markers of significance like # and * are not described in figure 2 which may confuse the readers. Further, please don’t show conclusions in figure legends. Please check all the figure legends to make sure they are suitable.
3. Please optimize the figure setting. For example, the text in Fig3A is deviated. Fig 2B lose some information.
4. Pay attention to the front, tense and grammar, I wish you can polish this article in English writing. Line 155, “in vitro” should be in italic; line 189, “ul” should be 1. The authors should check that all the labels of figures and tables in this article are correctly ordered. For example, in line 124, table 2 is not correct.
2. The description of Figure legends should be concise and pointed. For instance, The description of Figure 3 is too lengthy and to some extent similar to methods. What’s more, the markers of significance like # and * are not described in figure 2 which may confuse the readers. Further, please don’t show conclusions in figure legends. Please check all the figure legends to make sure they are suitable.
3. Please optimize the figure setting. For example, the text in Fig3A is deviated. Fig 2B lose some information.
4. Pay attention to the front, tense and grammar, I wish you can polish this article in English writing. Line 155, “in vitro” should be in italic; line 189, “ul” should be “μl”, line 244, “N-cadherine” is wrongly spelled. The tense is in chaos in this article.
5. It will be better to show the fluorescent images of fig1D in higher power field to see the colocalization of TET1 and N-cadherin more clear.
l”, line 244, “N-cadherine” is wrongly spelled. The tense is in chaos in this article.
5. It will be better to show the fluorescent images of fig1D in higher power field to see the colocalization of TET1 and N-cadherin more clear.

Experimental design

1. I was confused about the Fig1E, what do the numbers in X-axis and Y-axis mean? And the method for Fig1E should be clarified.
2. Are there any literatures reporting about the oxygen concentration of endometriosis? Why do you choose 5% O2 as condition.
3. The change from N-cadherin to Vimentin seems a bit unconvincing. Though N-cadherin and Vimentin are both markers for mesenchymal transition, from figure2B it seems the change tendency of vimentin is more similar to TET1. Maybe you can add more evidence of vimentin expression in endometriosis to make the story more reasonable.
4. You use the all endometrium and endometriosis tissue to extract the genomic DNA for 5-hmC analysis. I wondered how can it represent the glandular epithelial cells which are mainly studied in this article.
5.Why do you choose TET1 to study rather than TET2, TET3; which are also demethylation enzymes.

Validity of the findings

1. Why do you not choose student T test in your article?
2. From Fig4C, the TET1 regulated by HIF1a seems not so robust?
3. Please show raw data for histograms in figures.

Additional comments

Please extrude the highlight of this article in discussion and make the discussion more concise.

---

## Round 0.2 · accepted · Accept

I have the pleasure of announcing this good news to you. All three reviewers appreciated your efforts during the revision process. Congratulations!

Reviewer 1 ·

Basic reporting

There is still a scale bar in the middle of the image of Figure 2. JJ.

Experimental design

No comment

Validity of the findings

No comment

Additional comments

This reviewer found that the authors made significant changes and answered all of my concerns to satisfaction.

Reviewer 2 ·

Basic reporting

No comments.

Experimental design

No comments.

Validity of the findings

No comments.

Additional comments

The revised version of the manuscript and the rebuttal provided by the authors adequately address most comments and suggestions in the current version of the manuscript. Overall, it id appropriate for publication in PeerJ.

Reviewer 3 ·

Basic reporting

Tis article have improved its English expression.
The figures are OK.

Experimental design

The experimental design is OK.

Validity of the findings

The validity of findings is OK and the discussion is concise and logical.

Additional comments

Thank you for your careful revision. It's good.